🔓 | **Open Peer Review** | Environmental Microbiology | Research Article

# Active pathways of anaerobic methane oxidization in deep-sea cold seeps of the South China Sea

Qiuyun Jiang,[1,2] Hongmei Jing,[1,3,4] Xuegong Li,[1] Ye Wan,[1] I-Ming Chou,[1] Lijun Hou,[5] Hongpo Dong,[5] Yuhui Niu,[5] Dengzhou Gao[5]

**ABSTRACT**  Anaerobic oxidation of methane (AOM) is critical for controlling methane emissions from deep-sea cold seeps. Microbial groups associated with different AOM processes have been briefly investigated, but the AOM activities associated with the utilization of different electron acceptors in the deep-sea methane seeps remain largely unknown. Here, surface sediments (0–4 cm) from two cold seeps (Site F/Haima) and the Xisha trough in the South China Sea (SCS) were incubated to measure AOM potentials driven by nitrite, nitrate, and sulfate as different electron acceptors with isotopically labeled methane gas ($^{13}CH_4$), and the population shifts of microbial communities along the incubation were investigated by high-throughput sequencing and quantitative polymerase chain reaction based on both the 16S rRNA gene and functional genes. Nitrite and nitrate were preferentially used prior to sulfate during the incubation, and nitrate-/nitrite-dependent anaerobic methane oxidation (Nr-/N-DAMO) contributed more to the methane flux in both the cold seeps and the trough. The highest rates of DAMO and sulfate-dependent anaerobic methane oxidation (SAMO) were detected at Site F, consistent with the transcript abundance of *mcr*A and *dsr*B genes during the incubation, and might be explained by the *in situ* active functional groups. The former process might be explained by *Campylobacteria*, which could couple sulfide oxidation and nitrate consumption to further participate in the N-DAMO process, while the latter process might be due to the higher proportions of *Desulfobacterota*. This study elucidates the diversity and potential activities of major microbial groups in the DAMO and SAMO processes in the deep-sea cold seeps using isotopic tracing incubations and provides direct evidence for the occurrence of Nr-/N-DAMO as a previously overlooked microbial methane sink in the deep-sea hydrate-bearing sediments in the SCS.

**IMPORTANCE**  Cold seeps occur in continental margins worldwide and are deep-sea oases. Anaerobic oxidation of methane is an important microbial process in the cold seeps and plays an important role in regulating methane content. This study elucidates the diversity and potential activities of major microbial groups in dependent anaerobic methane oxidation and sulfate-dependent anaerobic methane oxidation processes and provides direct evidence for the occurrence of nitrate-/nitrite-dependent anaerobic methane oxidation (Nr-/N-DAMO) as a previously overlooked microbial methane sink in the hydrate-bearing sediments of the South China Sea. This study provides direct evidence for occurrence of Nr-/N-DAMO as an important methane sink in the deep-sea cold seeps.

**KEYWORDS**  cold seeps, anaerobic methane oxidation, isotopic tracing experiment, electron acceptors, $^{13}CH_4$

Cold seeps are formed by the expulsion of subsurface fluid into the seabed and are often rich in hydrogen sulfide, methane, and other hydrocarbons (1). Anaerobic oxidation of methane (AOM) is an important microbial process of methane cycling to

Address correspondence to Hongmei Jing, hmjing@idsse.ac.cn.

The authors declare no conflict of interest.

See the funding table on p. 15.

control the emission of methane from anoxic environments of the cold seeps and plays an important role in regulating methane content in the ocean (2, 3). In previous studies, AOM processes coupled to iron (4), manganese (4), and sulfate (5) reduction performed by anaerobic methanotrophic (ANME) archaea (ANME-1, ANME-2a/b/c, and ANME-3) have been demonstrated in marine methane seeps and hydrate layers. Sulfate-dependent anaerobic methane oxidation process (SAMO) was the most studied AOM process in the deep-sea methane seeps by far (6–8) with a consortium of ANME-1/2/3 and sulfate-reducing bacteria (SRB) of the genera *Desulfosarcina/Desulfococcus* or *Desulfobulbus* (9, 10), and could oxidize more than 80% of uprising methane in the sea floor on a global scale (11). More recently, the bacterium Candidatus *Methylomirabilis oxyfera* (*M. oxyfera*, NC10) can couple AOM to nitrite reduction through an intra-aerobic methane oxidation pathway (known as "nitrite-dependent AOM" or N-DAMO) (12), and a novel ANME lineage named Candidatus *Methanoperedens nitroreducens* (ANME-2d) population can perform nitrate-driven AOM pathway (known as "nitrate-dependent AOM" or Nr-DAMO) (13, 14) was revealed. These DAMO-related microbial groups have been identified in the sediments collected from mangroves (15, 16), freshwater sources (13, 17), the continental shelf, and cold seeps of the South China Sea (SCS) (18–20), suggesting the DAMO process might occur widely in aquatic environments. Nevertheless, the research on Nr-/N-DAMO in aquatic habitats is still very limited so far. For example, the community dynamics of DAMO-related microbes and their contribution to the AOM rate by utilization of different electron acceptors remain unexplored, and their methane removal potential as well as the underlying controlling mechanisms in aquatic environments remains unclear.

Sulfate ($SO_4^{2-}$), nitrite ($NO_2^-$), and nitrate ($NO_3^-$) are important electron acceptors for the anaerobic oxidation of methane in deep-sea sediments. $SO_4^{2-}$, a major constituent of seawater, has been known as the major electron acceptor for anaerobic oxidation of methane (21, 22) and preferentially occurs in deep oceanic waters (>800 m) (23). Comparatively, $NO_3^-$ (13) and $NO_2^-$ (12) as important oxidized compounds are thermodynamically more favorable than $SO_4^{2-}$ (24). Substrate utilization by these two processes and the major microbial groups involved have been investigated in various freshwater ecosystems and intertidal zones (23, 25, 26). Though SAMO was previously assumed to be the main AOM pathway in marine habitats (21, 27), a greater contribution of DAMO to methane sink was observed in the Zhoushan intertidal zone (25) and in the intertidal marsh soils of the Yangtze Estuary (26). This indicated that the contribution of these two processes varied in different marine habitats. By far, variation about the rate of the main AOM processes and the related functional groups among different deep-sea regions is still poorly known (28).

The hydrate-bearing deep-sea cold seep is a hot topic in the study of AOM processes. Various cold seeps are located in the northern part of the SCS (29), where the SAMO process is the most studied AOM process. The microbial communities (ANME and SRB) have been investigated (6, 7, 30), and rates of AOM and sulfate reduction were measured recently with $^{14}C$ or $^{35}S$ labeled tracers separately (31). In the last decade, microbial groups involved in the DAMO process were also identified based on the presence of 16S rRNA gene and functional genes (*pmo*A, *mcr*A, and *dsr*B) in the cold seeps and the Xisha trough in SCS (18–20). However, detection of the DAMO-related microbes at the DNA level does not guarantee the occurrence of this process, and the role of these microbial groups in the AOM processes needs to be further investigated. The lack of rates of AOM pathways coupling with different substrates of $SO_4^{2-}$ and $NO_2^-/NO_3^-$ represents a fundamental gap in our understanding of the dynamics of AOM processes and evaluation of their contributions in the deep-sea cold seeps.

Here, we conducted an incubation experiment using surface sediments collected from the two typical cold seeps and the Xisha trough of the SCS with $SO_4^{2-}$ and $NO_2^-/NO_3^-$ addition as electron acceptors. Community diversity and gene abundance of the microbial groups involved in the two major AOM processes, i.e., SAMO and DAMO, were investigated using high-throughput sequencing and quantitative polymerase chain

reaction (qPCR) based on 16S rRNA genes (bacteria and archaea) and functional genes (*mcr*A: ANME-2d subcluster, *pmo*A: NC10 bacteria, *dsr*B: SRB), and the potential AOM activity in the two processes was estimated with stable isotope tracing methods in order to elucidate the dynamics of the AOM process in the deep-sea hydrate-bearing sediments. This study provides novel insights into methane cycling in deep-sea cold seep ecosystems.

## MATERIALS AND METHODS

### Sample collection and chemical analysis

Surface sediment samples were collected from two cold seeps (Haima: 16°43′N, 110°28′E; Site F: 22°6′N, 119°17′E) and the Xisha trough (18°18′N, 114°08′E) located at the continent shelf of the SCS (Fig. 1) using a pushcore during June 2018 and June 2021, as described previously (19, 32). In total, nine stations were sampled with three stations in each region, i.e., SQ_56, SQ_58, and SQ_81 (from Haima), SQ_63, SQ_261, and SQ_264 (from Site F), and SQ_82, SQ_84, and SQ_87 (from Xisha trough). The top layer (0–4 cm) was sliced and divided into two parts: one immediately stored at 4°C for incubation experiments to determine the potential AOM rate and associated microbial groups, and the other stored at −80°C for physicochemical characterization.

Approximately 5 g of sediment was used for chemical analysis. The analysis of total organic carbon (TOC), total nitrogen (TN), total phosphate (TP), nitrate ($NO_3^-$), and ammonia ($NH_4^+$) was conducted at the Institute of Mountain Hazards and Environment, Chinese Academy of Sciences (Chengdu, Sichuan, China), according to Wang et al. (33). Briefly, $NO_3^-$ and $NH_4^+$ were detected with a colorimetric auto-analyzer (SEAL Analytical AutoAnalyzer 3, Germany) after 2 M KCl treatment. TOC and TN were determined by oven-drying the sediments at 105°C and then using an element analyzer (Elementar Vario Macro Cube, Germany). TP was measured with nitric-perchloric acid using the molybdate colorimetric method with a UV2450 (Shimadzu, Japan) after digestion of the sediment (34).

### Measurement of AOM rates

AOM activity was measured using a slurry experiment with the $^{13}CH_4$ isotopic-tracing technique. Briefly, about 2 g of sediments and 6 mL of $N_2$-purged deionized water

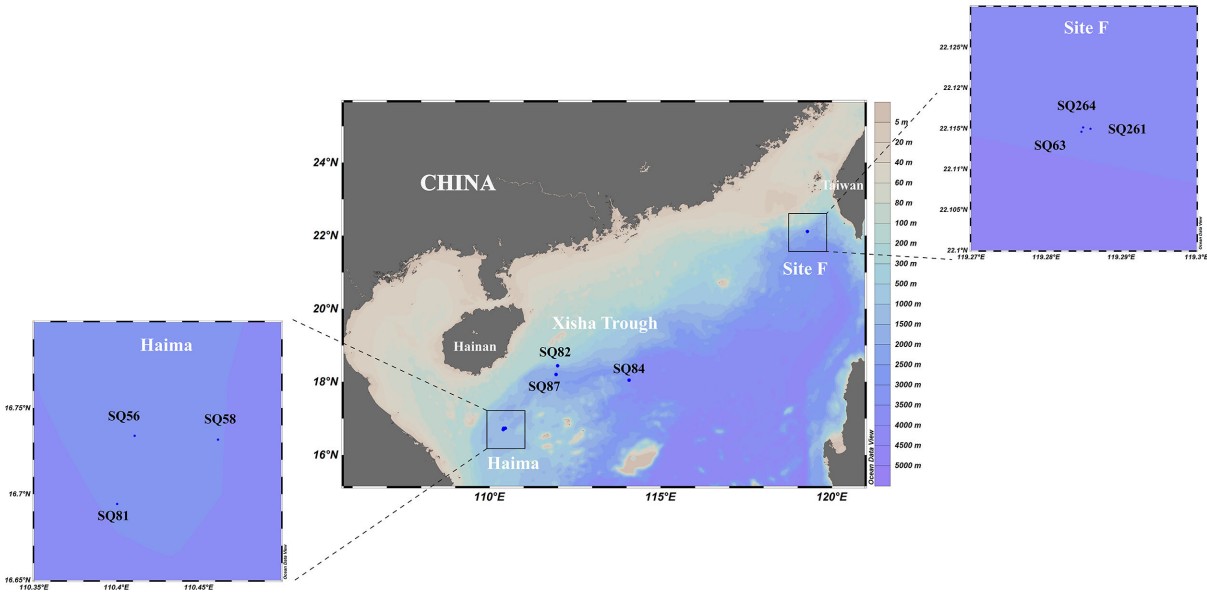

**FIG 1** Location of the sampling stations in the South China Sea. The map was plotted using Ocean Data View software.

(34.57‰) were transferred into 20 mL glass vials and then preincubated for 2 days (4°C, dark) on orbital shakers to remove background $NO_x^-$ ($NO_2^-$ + $NO_3^-$) and $SO_4^{2-}$. The slurries were subsequently split into different treatment groups: (i) $^{13}CH_4$ ($^{13}C$ at 99%), (ii) $^{13}CH_4$ + $NO_2^-$, (iii) $^{13}CH_4$ + $NO_3^-$, and (iv) $^{13}CH_4$ + $SO_4^{2-}$. Subsequently, the vials were injected with 50 µL of $N_2$-degassed stock solutions of either $NaNO_2$ (0.8 M), $NaNO_3$ (0.16 M), or 100 µL of $Na_2SO_4$ (2.4 M) for a final concentration of 5 mM $NaNO_2$, 1 mM $NaNO_3$, or 30 mM $Na_2SO_4$ in the slurry. After that, 2 mL of $^{13}CH_4$ was immediately injected into these vials in the headspace. All the slurries were incubated at *in situ* temperature (4°C) in the dark with gentle shaking at 120 rpm. The incubation time was set at 1 day (D1), 3 days (D3), 7 days (D7), and 14 days (D14). The total $CO_2$ concentration and production of $^{13}CO_2$ were measured on a gas chromatography (GC-2014, Shimadzu, Japan) and isotope ratio mass spectrometer (Delta V Advantage; Thermo Fisher Scientific, Germany), respectively. Potential rates of N-DAMO, Nr-DAMO, and SAMO were quantified according to the production of $^{13}CO_2$. The maximum methane oxidation rates were estimated from the increase in $CO_2$ between the start values and the highest observed values of $^{13}CO_2$ on the days of the experiments (23). Concentrations of $NO_2^-$, $NO_3^-$, and $SO_4^{2-}$ in the culture supernatant were determined using a spectrophotometric method (35, 36).

## RNA extraction, cDNA synthesis, and gene amplification

During the 14-day incubation, total RNA in the sediments (~0.5 g) at initial, D3 and D14 was extracted with the Soil RNA Kit (R6825-01; Omega Bio-Tek, Norcross, GA, USA), and cDNA was synthesized by the reverse transcription process using the SuperScript III First-Strand Synthesis SuperMix (Invitrogen, Carlsbad, CA, USA) and quantified by a NanoDrop-2000 spectrophotometer (Thermo Fisher Scientific, USA).

Using the cDNA as a template, the V3–V4 region of the bacterial 16S rRNA gene was amplified by PCR using the primers 338F (5′-ACTCCTACGGGAGGCAGCAG-3′) and 806R (5′-GGACTACHVGGGTWTCTAAT-3′) (37), while that of the archaeal 16S rRNA gene was amplified with primers 340F (5′-CCCTAYGGGGYGCASCAG-3′) and 806R (5′-GGAC-TACVSGGGTATCTAAT-3′) (38). Nested PCR was used for amplification of the *mcr*A gene of *Methanoperedens*-like archaea (ANME-2d) (39), the *pmo*A gene of *M. oxyfera*-like bacteria (NC10 bacteria) (40), and the *dsr*B gene of *Desulfosarcina*-like sulfate-reducing bacteria (DSRB) (41). The information on specific PCR primers for the above functional genes is listed in Table S1. PCR products were examined with SYBR-Safe-stained 1.2% agarose gels. The paired-end sequencing of all amplicons was then performed with an Illumina HiSeq PE250 sequencer (Novogene Co., Ltd., www.novogene.com).

## Sequence analysis

Sequences were quality-filtered, trimmed, de-noised, and merged using the DADA2 (v1.16) (42) plug-in of QIIME2 v2020.2 to obtain high-quality sequencing data, and the representative sequences were picked. The amplicon sequence variants (ASVs) table, which are analogs of the traditional OTUs, was filtered out by q2-filterfeature after removing the ASVs with frequencies less than 10. Resulting representative ASV sequences for functional genes (*mcr*A, *pmo*A, and *dsr*B) were used to calculate phylogenetic trees for subsequent analyses.

Taxonomic and compositional analyses of 16S rRNA (bacteria and archaea) were conducted using the q2-classifysklearn algorithm, and the SILVA (V.138) database was used as a reference with a threshold of 0.8. Annotations were obtained after removing contamination using the q2-feature-table plugin and visualized by the q2-taxa-barplot plugin. The ASVs annotated as mitochondria, chloroplasts, or eukaryotes were further removed. For the prediction of functional and metabolic profiles of the bacterial and archaeal communities based on the 16S rRNA gene sequences, the R package Tax4Fun (43) was used with the short reads mode disabled along with the SILVA database.

For the functional genes (*mcr*A, *pmo*A, and *dsr*B), phylogenetic analysis was performed separately using Mega X (44) with reference sequences retrieved from the GenBank database at NCBI (http://www.ncbi.nlm.nih.gov). The gene sequences were

aligned using the ClustalW algorithm. After testing the best substitution model, phylogenetic trees were constructed using the maximum-likelihood method with the substitution model (Tamura-Nei model for *mcr*A, Tamura 3-parameter model for *pmo*A and *dsr*B; bootstrap value of 1,000). Phylogenetic trees were visualized and edited through ITOL (V2.0) (45).

## Quantitative PCR

Abundance of the functional genes (*mcr*A, *pmo*A, and *dsr*B) was quantified using the Bio-Rad System (Bio-Rad Inc., USA) and TB Green Premix Ex Taq II (Takara Bio Inc., Shiga, Japan) with primers AAA641F/AAA834R (39), qP1F/qP2R (46), and DSRB-213f/DSRB-658r (47) using cDNA as template, respectively. The abundance of bacterial and archaeal 16S rRNA genes was quantified with the same primer sets as for the amplicon sequencing mentioned above. The specific information for all the qPCR primers is listed in Table S1. The total volume of qPCR mixtures was 20 µL, consisting of 10 µL TB Green Premix, 0.4 µL ROX, 0.8 µL forward/reverse primers, 6.0 µL double distilled water, and 2 µL of template cDNA. The annealing temperature of AAA641F/AAA834R and DSRB-213f/DSRB-658r was 60°C, while that of qP1F/qP2R was 57°C. Standard curves were constructed using a series of tenfold dilutions of the standard plasmids (known copy numbers) containing the targeted genes (48). Triplicate qPCR reactions were performed for each sample with double-distilled water as a negative control, and the gene copy number was normalized to the quantity of the gene.

## Statistical analysis

The nonlinear regression was used to simulate the relationship between the concentration of $NO_2^-$, $NO_3^-$, and $SO_4^{2-}$ in the culture supernatant and $CO_2$ production at the end of incubation using the ggpmisc extension to the ggplot2 package in R (49). Pearson's correlation coefficients were calculated to assess correlations between electron acceptors and $CO_2$ production. Beta diversity was generated with QIIME 2 using q2-diversity and visualized using non-metric multidimensional scaling (nMDS) plots. ANOSIM (analysis of similarities) was used to analyze the similarities of the microbial community compositions among different regions.

## RESULTS

### Geochemical characterization of the sediments

The Xisha trough was geographically closer to the Haima (Fig. 1), with similar nutrient levels. Among the different parameters measured, the highest TN (0.09–0.17%), TOC (0.67–1.27%), and TS (684.80–3,017.88 mg/kg) contents were found in the Haima (Table 1) and were significantly higher than those at Site F ($P < 0.01$). The carbon to nitrogen molar ratios (C/N) (6.87–8.14), $NH_4^+$ (9.49–15.84 mg/kg), $NO_3^-$ (1.01–5.84 mg/kg), and TP (477.01–918.97 mg/kg, $P < 0.05$) were the highest at Site F compared with Haima and Xisha.

### Potential AOM rates and consumption of different electron acceptors

The $^{13}CH_4$ tracing experiments showed that production of $^{13}CO_2$ in all sediments amended with different electron acceptors ($NO_2^-$, $NO_3^-$, and $SO_4^{2-}$) was always higher in the cold seeps than that in the trough during the 14-day incubation period (Fig. 2; Fig. S1). In detail, the $CO_2$ production was the highest at Site F [6.20 nmol g$^{-1}$ (wet sediment); Fig. 2D through F], followed by Haima [3.69 nmol g$^{-1}$ (wet sediment); Fig. 2A through C], and the lowest in the Xisha trough [3.53 nmol g$^{-1}$ (wet sediment); Fig. 2G through I] after the 14-day incubation. The $CO_2$ production of DAMO (Nr-/N-DAMO) increased rapidly during the first seven days, while that of SAMO increased rapidly from D3 to D14 (Fig. 2). The $CO_2$ production of the DAMO process (Nr-/N-DAMO) in the cold seeps was higher than that in the trough, with the highest value appearing at Site F. The highest

**TABLE 1** Geochemical information of sediment samples collected from cold seeps in the South China Sea

| Regions | Station | Longitude (°E) | Latitude (°N) | Depth (m) | TN (%) | TOC (%) | TP (mg/kg) | TS (mg/kg) | C/N Ratio | NH$_4^+$ (mg/kg) | NO$_3^-$ (mg/kg) |
|---|---|---|---|---|---|---|---|---|---|---|---|
| Haima | SQ58 | 110.46 | 16.73 | 1,388 | 0.17 | 1.27 | 678.28 | 1,468.90 | 7.66 | 11.86 | 1.23 |
| | SQ79 | 110.41 | 16.73 | 1,377 | 0.15 | 1.15 | 531.86 | 3,017.88 | 7.87 | 10.46 | 1.02 |
| | SQ81 | 110.40 | 16.69 | 1,366 | 0.16 | 1.18 | 477.01 | 1,574.94 | 7.41 | 9.49 | 1.05 |
| Site F | SQ63 | 119.28 | 22.11 | 1,165 | 0.10 | 0.75 | 918.97 | 137.42 | 7.84 | 15.84 | 1.06 |
| | SQ261 | 119.28 | 22.11 | 1,128 | 0.09 | 0.69 | 758.29 | 684.80 | 7.70 | 14.37 | 5.84 |
| | SQ264 | 119.28 | 22.12 | 1,128 | 0.10 | 0.82 | 797.97 | 115.61 | 8.14 | 9.73 | 1.50 |
| Xisha trough | SQ82 | 111.99 | 18.44 | 1,732 | 0.14 | 1.11 | 610.99 | 1,127.57 | 8.00 | 13.11 | 1.01 |
| | SQ84 | 114.08 | 18.05 | 3,408 | 0.10 | 0.67 | 555.34 | 925.19 | 6.87 | 11.63 | 1.22 |
| | SQ87 | 111.94 | 18.20 | 2,200 | 0.12 | 0.91 | 659.45 | 904.08 | 7.56 | 9.80 | 1.41 |

$CO_2$ production of SAMO was observed at Site F, while there was no obvious difference between Haima and Xisha troughs (Fig. 2; Fig. S1).

The production of $^{13}CO_2$ increased rapidly from D3 to D7 during the incubation with various electron acceptors addition (Fig. 2); therefore, the AOM rates were estimated using values during this period. The AOM rate observed at Site F [0.25 nmol g$^{-1}$ (wet sediment) d$^{-1}$] was the highest, followed by the Haima [0.13 nmol g$^{-1}$ (wet sediment) d$^{-1}$], and the Xisha trough [0.08 nmol g$^{-1}$ (wet sediment) d$^{-1}$] (Fig. 3A and B). On average, AOM rates in the cold seeps [0.06–0.34 nmol g$^{-1}$ (wet sediment) d$^{-1}$] was higher than those in the trough [0.02–0.12 nmol g$^{-1}$ (wet sediment) d$^{-1}$]. Specifically, the rate of DAMO (Nr-/N-DAMO) in the cold seeps was higher than that in the trough, especially at Site F [0.34 and 0.31 nmol g$^{-1}$ (wet sediment) d$^{-1}$, respectively] (Fig. 3B). The rate of SAMO was the highest at Site F [0.21 nmol g$^{-1}$ (wet sediment) d$^{-1}$], followed by the Xisha trough

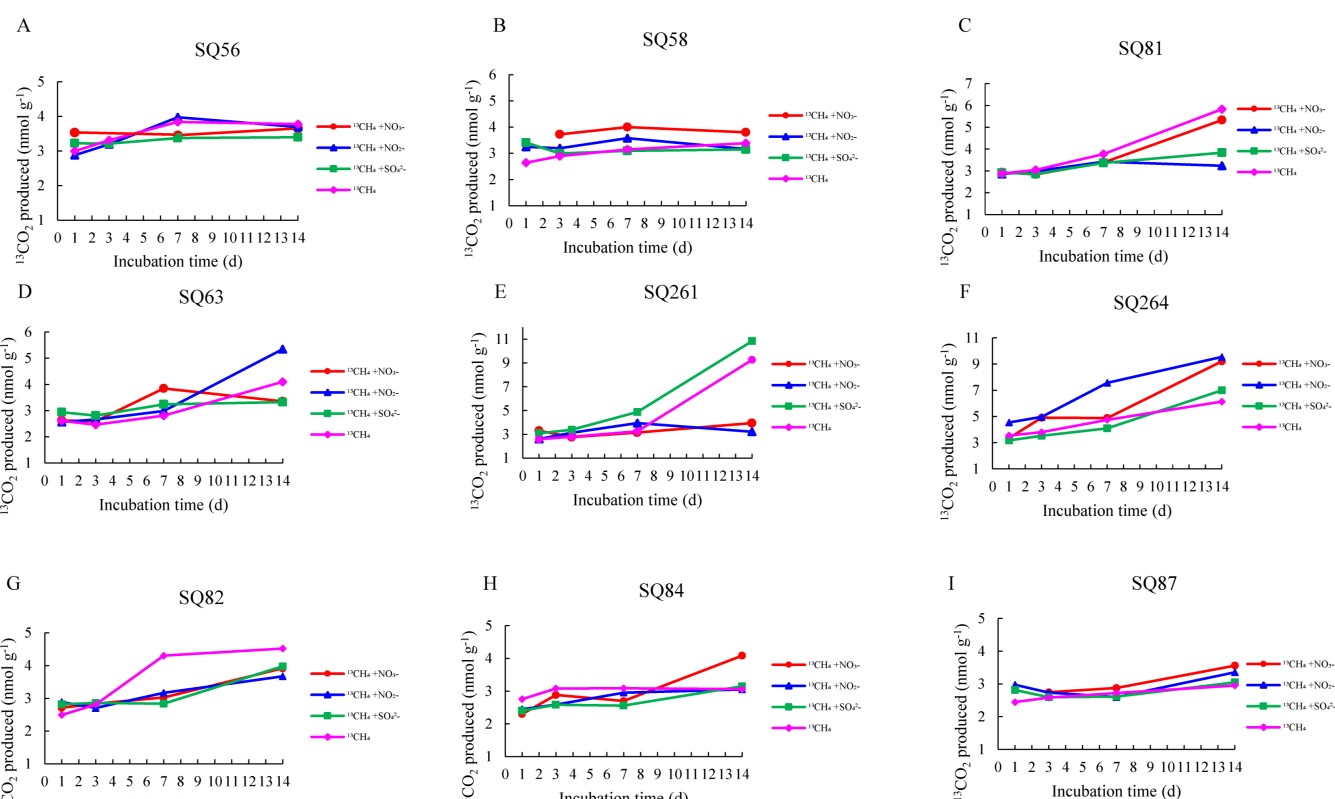

**FIG 2** $^{13}CO_2$ production in slurries of sediment amended with $^{13}CH_4$ and different electron acceptors during 14-day incubations. Haima: A–C; Site F: D–E; Xisha trough: G–I.

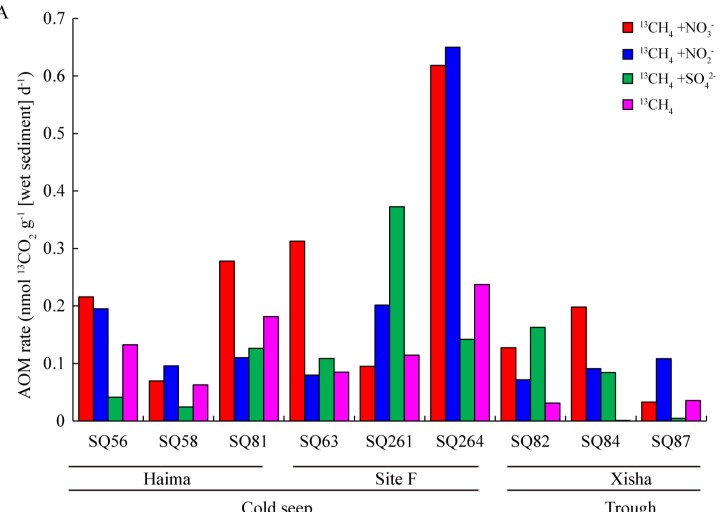
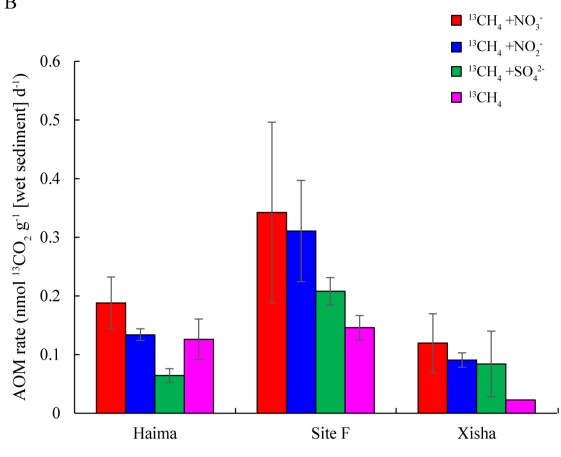

**FIG 3** Spatial variations of the potential AOM rate in different samples (A) and regions (B). All the rates were calculated based on wet sediment weight. Error bars are the standard error of three replicates. Error bars are the standard error of three replicates.

[0.08 nmol g$^{-1}$ (wet sediment) d$^{-1}$], and the lowest in the Haima [0.06 nmol g$^{-1}$ (wet sediment) d$^{-1}$)] (Fig. 3B).

In general, obvious consumption of nitrite appeared after 3 days of incubation (Fig. 4A), i.e., the nitrite content in the cold seeps was lower than that in the trough, especially in the Haima cold seep with nitrite almost exhausted at D3. After 14-day incubation, the nitrate content decreased significantly in both the cold seeps and the trough, especially at Site F (Fig. 4B). As for sulfate, its concentration changed slightly from 33.67mM to 25.65 mM in the trough during the 14-day incubation, and was barely consumed in the cold seeps (Fig. 4C). Pearson's correlation coefficients demonstrated that a significant correlation existed between the concentration of nitrate in the culture supernatant and $^{13}CO_2$ production during the whole incubation ($P < 0.05$; Table S3). Nonlinear fitting showed that nitrite (Fig. 4D), nitrate (Fig. 4E), and sulfate (Fig. 4F) in the culture supernatant were consumed continuously with the production of $^{13}CO_2$ in all the treatment groups, with nitrite being consumed first, followed by nitrate and sulfate.

## Abundance of gene transcripts

In all the samples, the abundance of bacterial 16S rRNA gene transcripts (Fig. 5A) was generally two orders of magnitude higher than that of archaea (Fig. 5B). The abundance of the 16S rRNA gene and functional genes of *mcr*A (Fig. 6A) and *dsr*B (Fig. 6C) was significantly higher in the cold seep than that in the trough, especially at Site F ($P < 0.01$), while that of the NC10 gene (*pmo*A) was less variable among different samples (Fig. 6B).

During the incubation with electron acceptors added, both the bacterial and archaeal 16S rRNA gene abundance increased during the first 3 days, then decreased at the end of incubation in Haima. They had little variation among different groups at Site F; in contrast, they decreased firstly after 3-day cultivation and then increased gradually but still less than the initial level in the Xisha trough. During the period of D3 to D14, the gene abundance of *mcr*A decreased in the Haima and Xisha but increased at Site F with the addition of NO$_3^-$ (Fig. 6A); that of NC10 gene (*pmo*A) was less variable in the Haima and Xisha but increased at Site F with the addition of NO$_2^-$ (Fig. 6B). However, the gene abundance of *dsr*B was less variable in all samples with the addition of SO$_4^{2-}$ (Fig. 6C).

## Diversity of communities and predicted functions

For the prokaryotic communities, *Proteobacteria,* comprised mainly of α- and γ-Proteobacteria, was the predominant bacterial phylum, especially in Haima (Fig. 7A).

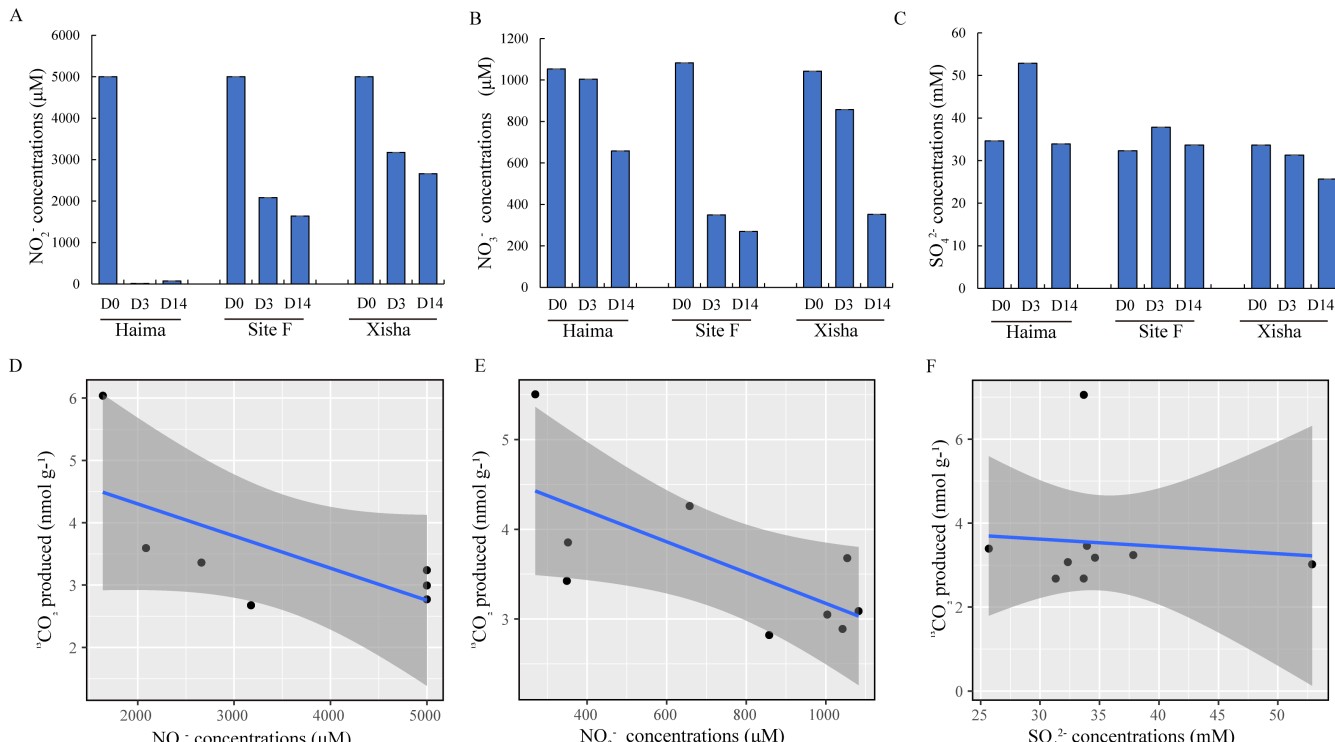

**FIG 4** Concentrations of $NO_2^-$ (A), $NO_3^-$ (B), and $SO_4^{2-}$ (C) in culture supernatant with different incubation times; nonlinear fitting between the concentrations of $NO_2^-$ (D), $NO_3^-$ (E), and $SO_4^{2-}$ (F) in culture supernatant and $CO_2$ production at the end of incubation. Error bars are the standard error of three replicates. Error bars are the standard error of three replicates.

The relative abundance of *Campylobacteria* (formerly called *Epsilonproteobacteria*) and *Desulfobacterota* was the highest at Site F ($P < 0.05$), while that of *Actinobacteria* and *Methylomirabilia* (NC10) was the highest in the Haima and the Xisha, respectively (Fig. 7A). *Halobacterota* was the predominant archaeal phylum in cold seeps, especially at Site F ($P < 0.01$), while *Crenarchaeota* was predominant in the Xisha trough ($P < 0.01$) (Fig. 7B). *Euryarchaeota* and *Nanoarchaeota* occupied higher proportions in the Xisha trough than the cold seeps ($P < 0.05$) (Fig. 7B). NMDS analysis of bacterial and archaeal communities showed that three distinct clusters were formed, corresponding to the three sampling regions (Fig. 7C and D).

For functional prediction of total bacterial ASVs, methanogenesis, especially the acetoclastic methanogenesis, was the major pathway in the cold seeps (Fig. 7E), and the relative abundance of this pathway and $CO_2$ methanogenesis decreased in all treatments at the end of incubation, while that of dissimilatory nitrate reduction was more abundant in the trough than the cold seeps and increased with assimilatory sulfate reduction after incubation in all the treatments (Fig. 7E). For functional prediction of total archaeal ASVs, the methanogenesis pathway was the major one in the cold seeps, especially at Site F, where there was a higher abundance of methanol, acetoclastic, and $CO_2$ methanogenesis pathways (Fig. 7F). The proportion of dissimilatory nitrate reduction was higher in the trough than the cold seeps and increased after incubation in all treatments (Fig. 7F).

## Phylogeny of functional genes

Four distinct clusters, i.e., ANME-2d, ANME-2e, *Methanosarcinaceae*, and *Methanocellales*, were observed in the phylogenetic tree based on the top 30 ASVs of the *mcr*A gene (Fig. S2). ANME-2e and *Methanocellales* were the two major clusters. The highest abundance of *mcr*A ASVs was in the $^{13}CH_4 + NO_3^-$ treatment groups of Haima and Xisha trough. The ASVs abundance of ANME-2d was higher in the $^{13}CH_4 + NO_3^-$ treatment groups of Haima

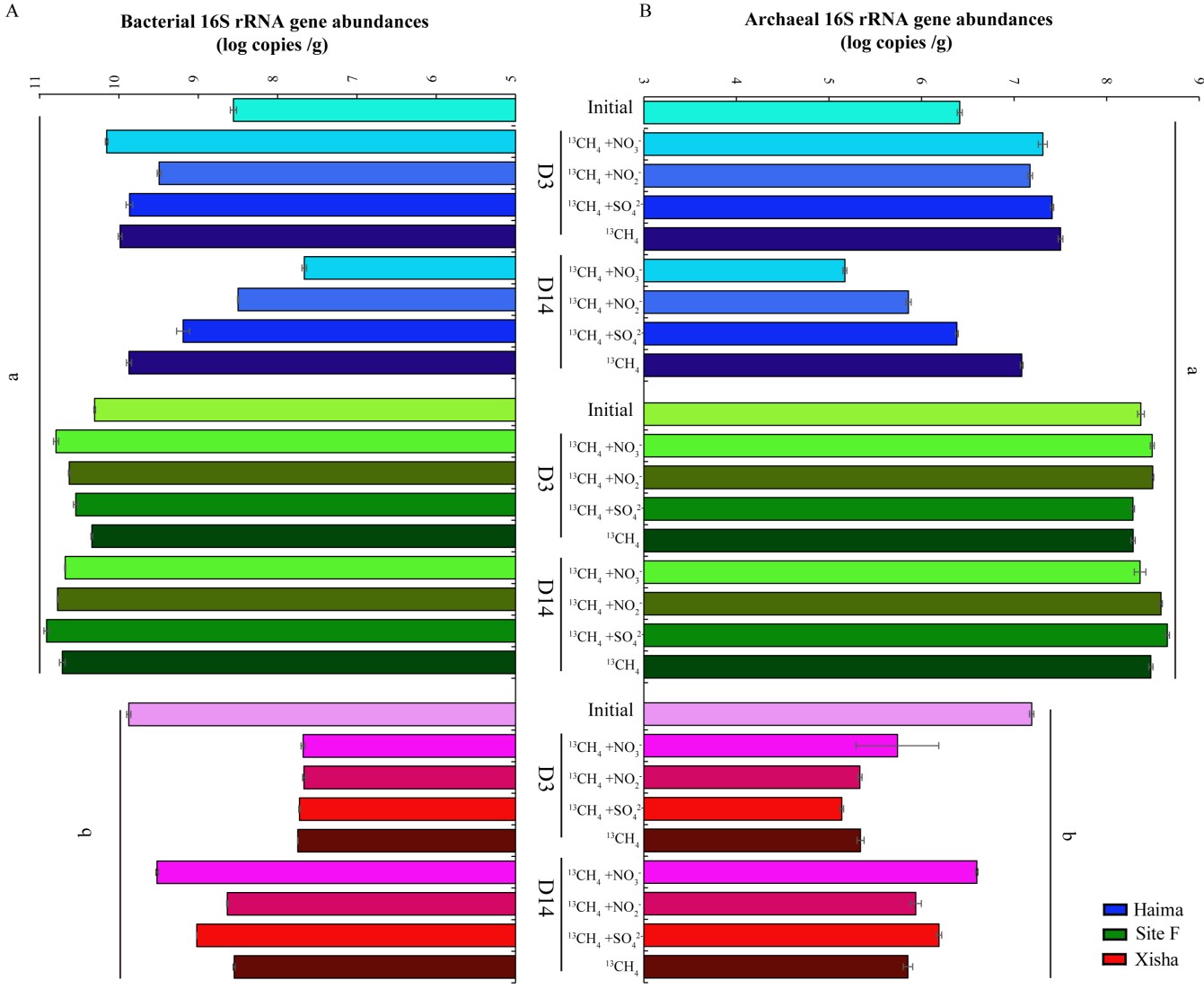

**FIG 5** Abundance of bacterial (A) and archaeal (B) 16S rRNA gene transcripts (log copies in per gram of wet sediment) with different substrates at the initial and end of incubation. ab: $P < 0.01$ (one-way analysis of variance). Error bars are the standard error of three replicates. Error bars are the standard error of three replicates.

and Xisha trough. Top 30 ASVs of the *pmo*A gene were used to construct a phylogenetic tree, with two distinct clusters formed (Fig. S3). Cluster I contained most of the ASVs (23 of 30 ASVs). The highest abundance of *pmo*A ASVs was at Site F, especially in the $^{13}CH_4 + NO_2^-$ treatment groups, followed by the Haima and Xisha troughs (Fig. S3). Top 20 ASVs of the *dsr*B gene were used to construct a phylogenetic tree and fell into three distinct clusters, i.e., *Syntrophobacteraceae*, *Desulfobacteraceae*, and Group IV (Fig. S4). *Syntrophobacteraceae* and Group IV were the two major clusters (Fig. S4). The highest abundance of *dsr*B ASVs was in the $^{13}CH_4 + SO_4^{2-}$ treatment groups in the Xisha (Fig. S4).

## DISCUSSION

### Potential AOM rates in the cold seeps and trough

The current work represents the first empirical assessment of the co-occurrence and potential role of DAMO and SAMO microbes in the cold seeps of SCS, using molecular and $^{13}CH_4$ tracer experiments. After the addition of different electron acceptors, the AOM (DAMO and SAMO) rate was apparently higher in the cold seeps, especially at Site F,

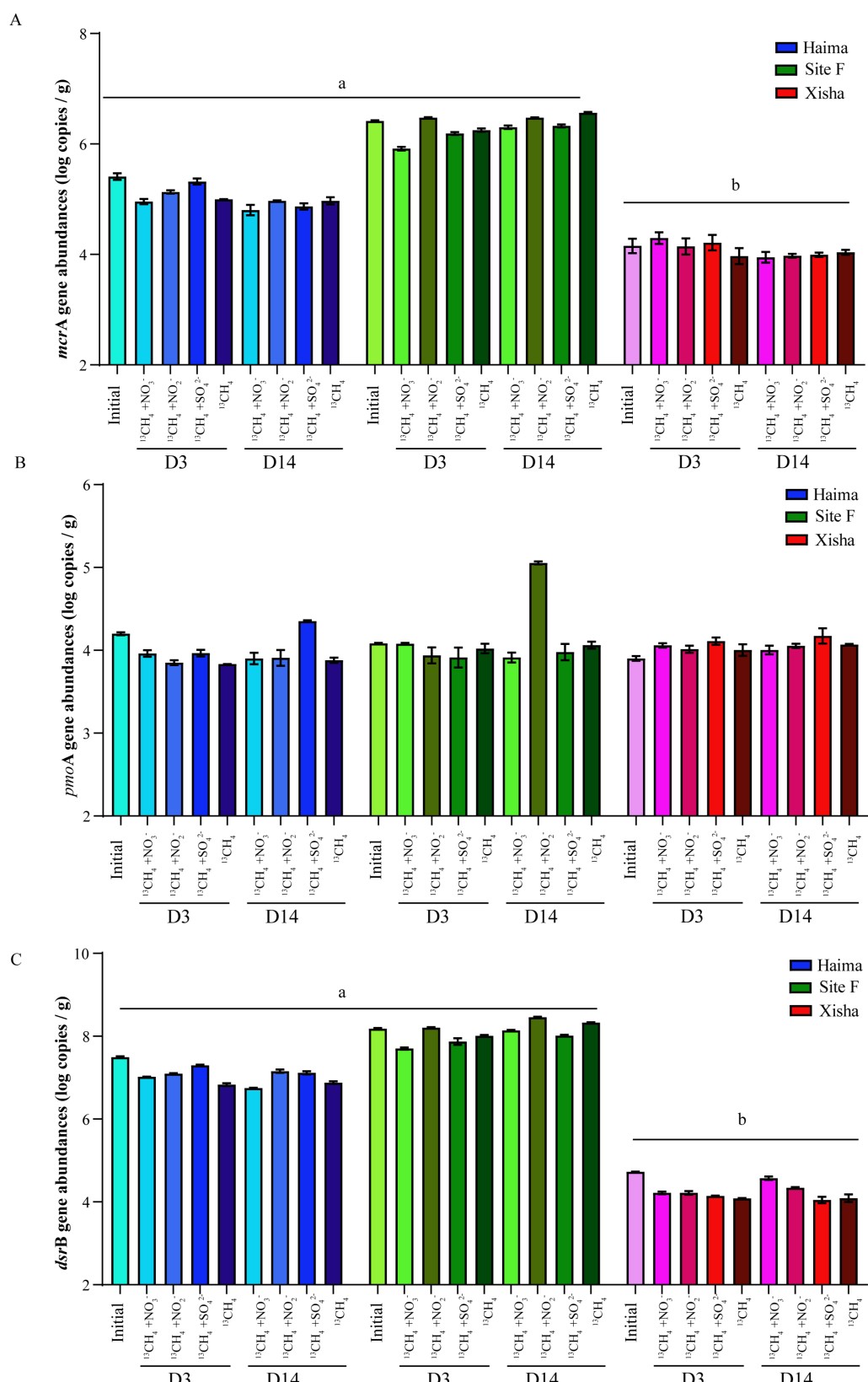

**FIG 6** Abundance of different functional gene transcripts (log copies per gram of wet sediment) at the initial and end of incubation. *mcr*A: (A), *pmo*A: (B), and *dsr*B: (C). ab: *P* < 0.01 (one-way analysis of variance). Error bars are the standard error of three replicates. Error bars are the standard error of three replicates.

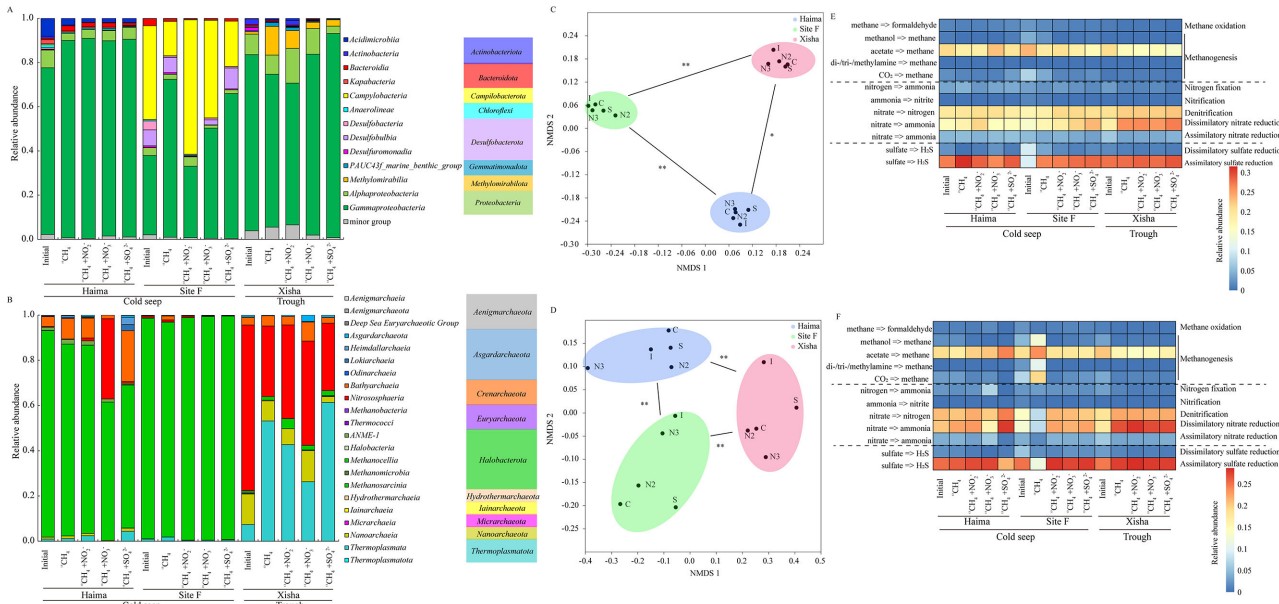

**FIG 7** Community composition of bacteria (A) and archaea (B) at the class level; non-metric multidimensional scaling (NMDS) plots of bacterial (C) and archaeal (D) communities. [**: $P < 0.01$; *: $P < 0.05$ (one-way analysis of variance)]; heatmaps of major predicted pathways in the methane, nitrogen, and sulfur cycles for bacteria (E) and archaea (F). I: initial, C: $^{13}CH_4$, N2: $^{13}CH_4 + NO_2^-$, N3: $^{13}CH_4 + NO_3^-$, S: $^{13}CH_4 + SO_4^{2-}$.

than in the Xisha trough. This might be related to the status of methane and the activity of the seeps. Bubbling of fluids from the two cold seeps was observed during diving with the deep-sea HOV on the cruise, although no *in situ* methane concentration was measured during the sampling. Haima has exhibited a decline in activity in recent years (50), while Site F is currently active (29) with a high concentration of methane gas (51), and the Xisha trough with large amount of gas hydrate without cold seeps formed (52). In addition to the electron acceptors, the AOM process could be controlled by the diversity and composition of microbial groups originally living in the different samples, which was selectively supported by the concentrations and physical forms of methane. Distinct microbial communities formed in the three different regions; therefore, it was reasonable to assume that the methane availability in the three regions was different, which subsequently affected the microbial communities and associated AOM processes.

SAMO processes have been reported in all the cold seeps and trough of the SCS, in which the microbial communities (ANME and SRB) have been investigated (6, 7, 30), and rates of AOM and sulfate reduction were measured recently (31). The rate of SAMO was the highest at Site F, suggesting that the microbial communities involved in SAMO might be active, and this was consistent with the abundant presence of the *dsr*B gene transcripts at Site F. In addition, $SO_4^{2-}$ also stimulated AOM in the cold seeps and the trough, suggesting that high methane concentrations support a greater diversity of AOM pathways. A similar phenomenon has also been observed in the pockmark (8) and Thuwal cold seeps (53), with the potential influence of sulfate concentrations (54). Highly diversified substrates utilizing capabilities of different SRB groups found in our study reflected a better adaptability to the deep-sea environment (55), and a potentially higher ecological contribution to AOM progress in the cold seeps and trough (20). Though SAMO was the main $CH_4$ removal pathway in marine habitats (21), a greater contribution of DAMO to $CH_4$ sink was reported in the Zhoushan intertidal zone (25). Here, we also found lower potential SAMO rates than that of DAMO in the deep-sea cold seeps, especially at Site F, indicating that the DAMO process was an important, widely occurring $CH_4$ sink previously overlooked in deep-sea cold seeps. In the process of electron acceptor utilization in this study, obvious consumption of nitrite and nitrate appeared at respective levels of D3 and D14, while that of sulfate varied slightly during

the whole incubation, suggesting that nitrite and nitrate were preferentially used prior to sulfate. The $CO_2$ production of DAMO (N-/Nr-DAMO) increased rapidly at D7, while that of SAMO increased rapidly at D14, further providing evidence for preferential utilization of electron acceptors. Nitrite and nitrate are thermodynamically more favorable electron acceptors than sulfate, and $NO_2^-$/$NO_3^-$-dependent AOM is energetically more favorable (56). Thus, these denitrifying methanotrophs may have an advantage over the sulfate-dependent methanotrophs in consuming $CH_4$ in the hydrate-bearing deep-sea cold seeps and the trough.

N-/Nr-DAMO is a recently discovered process of anaerobic oxidation of $CH_4$ using $NO_2^-$/$NO_3^-$ as electron acceptors (12, 13). This process is an important link between the two major global nutrient cycles of carbon and nitrogen (25), as well as the anaerobic ammonium oxidation (anammox) process, which also interconnects both cycles (57). The potential N-/Nr-DAMO activity measured in this study was higher in the cold seeps, indicating that the microbial groups involved in these two processes may be more active in the cold seeps than the trough, especially at Site F. N/Nr-DAMO rates at Site F were similar to those reported from Zhoushan Island (25), but slightly lower than those reported from the intertidal wetland of the Yangtze Estuary (26). These sediments of islands and estuaries are known to support high *in situ* rates of nitrate reduction, which are characterized by stable environmental conditions and the coexistence of methane and nitrate/nitrite (25, 26). The high nitrate reduction delivering nitrite to further drive the DAMO process (58) might be occurred at Site F, where a higher concentration of $NO_3^-$ would possibly support a higher *in situ* rate of nitrate reduction.

Based on the average N-DAMO rate [ranging from 0.089 to 1.093 nmol $^{13}CO_2$ $g^{-1}$ (dry sediment) $d^{-1}$; Fig. 8], the average density of sediment (2.60 g $cm^{-3}$) (59), and the area of hydrate-bearing sediments in the SCS (approximately equivalent to $3.275 \times 10^4$ $km^2$) (60), we estimate the amount of $CH_4$ oxidation according to Niu et al. (61), approximately a total of 2.39–4.99 Gg $yr^{-1}$ $CH_4$ at the Haima, 3.42–23.09 Gg $yr^{-1}$ $CH_4$ at Site F, and 1.88–2.82 Gg $yr^{-1}$ $CH_4$ at the Xisha trough could be oxidized to $CO_2$. However, the estimation of the amount of methane being oxidized in those sediments still appears largely uncertain because this process is likely to be highly variable in space and time owing to spatial and temporal heterogeneity in substrate availability and redox potential. In addition, compared with the *in situ* condition, the excessive addition of electron acceptors for incubation in this study might result in an overestimation of the amount of methane oxidation occurring in the sediments. Thus, more studies that mimic the *in situ* conditions as closely as possible are required to determine the quantitative importance of methane oxidation in the hydrate-bearing sediments of the SCS.

## Phylogeny and abundance of functional groups

Based on the *mcr*A gene, four distinct clusters formed, with the methanogenic archaeon *Methanocellales* accounting for the highest proportion. Although the primers used in this study had by far the best amplification specificity for ANME-2d, *Methanocellales* and other ANME clades could be amplified as well. This phenomenon of mismatch has been reported in previous studies (26). The *Methanocellales* cluster was closely related to sequences recovered from rice roots (62) and oil sludge (63). The ANME-2d cluster had the highest abundance in the $^{13}CH_4$ + $NO_3^-$ treatment groups of the Haima and the Xisha troughs, indicating that the Nr-DAMO process was facilitated by nitrate addition in the deep-sea cold seep and trough. This cluster was closely related to sequences recovered from Yangtze Estuary sediments (26) and Indonesian river sediments (39), which are typical habitats for the occurrence of Nr-DAMO processes (13). As for the *pmo*A gene, two distinct clusters formed, with Cluster I closely related to sequences recovered from the sediments of the SCS (18, 64), in which NC10 bacteria have been identified (18, 19) and their associated N-DAMO process has been proposed as well (20). The relative abundance of Cluster I was higher in the $^{13}CH_4$ + $NO_2^-$ treatment groups of the cold seeps than the trough. Different dominant ASVs existed in different samples, suggesting that different genotypes of DAMO bacteria were favored by different

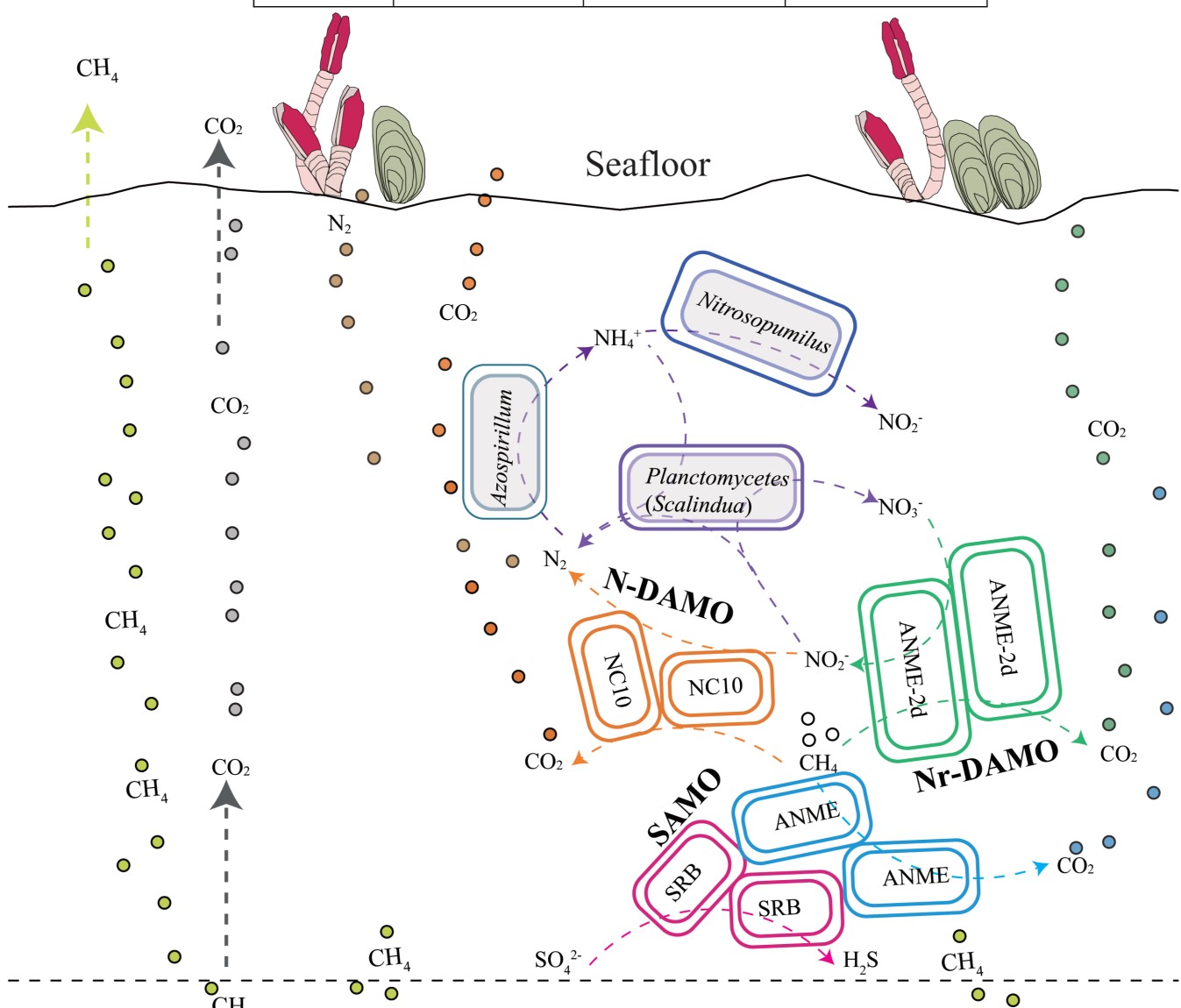

| Deep-sea cold seeps | | | |
|---|---|---|---|
| | Nr-DAMO | N-DAMO | SAMO |
| Haima | 0.081-0.332 | 0.113-0.236 | 0.029-0.151 |
| Site F | 0.165-1.039 | 0.162-1.093 | 0.222-0.645 |
| Xisha | 0.040-0.239 | 0.089-0.133 | 0.006-0.203 |

**FIG 8** Abstract art for $CH_4$ oxidation potentials of microbes in cold seeps and troughs. The unit of AOM rates is nmol g$^{-1}$ (dry sediment) d$^{-1}$.

environmental conditions, which was consistent with the previous finding that most of the habitats contained one certain *pmo*A ASV (65). In addition, among the three distinct clusters formed for the *dsr*B gene, clusters *Syntrophobacteraceae* and Group IV were closely related to sequences from deep-sea sediments of Nankai Trough (66), while that of *Desulfobacteraceae* was closely related to sequences from sediments of Changjiang Estuary (67).

Dynamics of gene transcripts for different functional genes were measured to reflect the active microbial groups during the incubation with different electron acceptors added, since detection of microbial groups at the DNA level does not reflect

the occurrence of biogeochemical processes and metabolic active groups. Generally, transcripts of the *mcr*A and *dsr*B genes were more abundant in the cold seeps than in the trough, while those of NC10 (*pmo*A) varied slightly in different regions. Considering the unspecificity of the *mcr*A primer, this might suggest that the first two functional groups were selectively favored by the enriched substrates in the cold seeps (6, 30, 68), while the N-DAMO process was almost equally important in the hydrate-bearing cold seep and trough (20). During the incubation period of D3 to D14, the transcripts of *mcr*A, *pmo*A, and *dsr*B genes at Site F increased in the treatment groups with $NO_3^-$, $NO_2^-$, and $SO_4^{2-}$ addition as electron acceptors, respectively. It was consistent with the highest AOM (DAMO and SAMO) rate at Site F measured during the incubation, suggesting that the AOM (DAMO and SAMO) process was highly active at Site F.

## Community similarity and impacting parameters

Distinct prokaryotic communities were formed in the three hydrate-bearing regions, respectively. This clear distribution pattern might be attributed to the *in situ* geochemical conditions in the sediments, especially TN, TOC, $NO_3^-$, and TS (20). The importance of those parameters in driving the spatial distribution of microbial communities in cold-seep ecosystems has been reported previously (6, 20, 69). In addition, concentrations and physical forms of methane in those different regions could be different as well, as mentioned above (29, 51, 53), subsequently supporting distinct prokaryotic and functional communities.

In addition to the predominant *Proteobacteria*, sulfate-reducing bacteria (*Desulfobulbus*, *Desulfococcus,* and *Desulfobacteraceae*) were found in the cold seeps and the trough (Table S2), consistent with previous studies in these regions (20). They were more abundant in all treatment groups and are known to be direct syntrophic partners of ANME-1/2/3 for the anaerobic oxidation of methane, suggesting a potential occurrence of the SAMO pathway in deep-sea habitats. NC10 bacteria were also detected in all studied regions, with a higher proportion in the trough, in consistence with their distribution patterns reported previously (18–20). *Campylobacteria* had a high proportion at Site F and has been found in a variety of sulfur-containing environments, including cold seeps (70, 71) and hydrothermal vents (72). This microbial group had the capability of coupling sulfide oxidation with $O_2$/nitrate consumption to participate in the rTCA cycle (73); nitrate, in turn, is reduced to gaseous nitrogen, possibly participating in the N-DAMO process, and possibly to ammonia (74). Hence, although the abundance of *pmo*A gene transcripts was high in both the cold seeps and the trough, the DAMO rate was extremely high at Site F, in which *Campylobacteria* might play an important role through involvement in nitrate reduction.

As for archaea, *Halobacterota* was most abundant in all samples of the cold seeps, suggesting a potentially high level of methane production occurred because this microbial group has a more diverse substrate range, including hydrogenotrophic, aceticlastic, and methylotrophic members. *Methanosarcinales* was the predominant *Halobacterota* in the cold seeps. In addition to $H_2/CO_2$, acetate, and common methyl-containing compounds, some *Methanosarcinales* species can also use more exotic substrates (75). Ammonia-oxidizing archaea *Nitrososphaeria*, which belonged to *Crenarchaeota*, was predominant in the Xisha trough. This group oxidized ammonia into $NO_2^-$ and $NO_3^-$, which might provide substrates for the DAMO process. In addition, the higher proportions and transcript abundance of *Methanoperedenaceae* (ANME-2d) in the cold seeps provided extra evidence for the occurrence of the Nr-DAMO process. Similarly, higher proportions of ANME groups (ANME-1a, ANME-1b, ANME-2a/2b, ANME-2c, and ANME-3) and transcripts of *dsr*B in the cold seeps further supported the importance of the SAMO process in the deep-sea cold seeps.

In summary, we provide direct evidence for the occurrence of Nr-DAMO and N-DAMO processes as a previously overlooked methane sink and the first evidence of multiple active pathways of AOM (DAMO and SAMO) driven by different electron acceptors in the hydrate-bearing sediments of the SCS. Among these, nitrite- and nitrate-dependent

AOM played an important role in regulating methane emissions in the studied regions harboring different concentrations and physical forms of methane together with varied microbial communities, which suggested a tight link between the diversity of AOM pathways and the availability of methane and associated active functional groups.

## ACKNOWLEDGMENTS

The authors thank the pilots of the deep-sea HOV "Shen Hai Yong Shi" and the crew of the R/V "Tan Suo Yi Hao" for their professional service during the cruise of TS07 in June 2018, and the crew of the R/V "Ke Xue Hao" for their professional service during the cruise in June 2021.

This work was supported by the National Key R&D Program of China (2022YFC2805505 and 2022YFC2805304) and the Hainan Province Science and Technology Special Fund (ZDKJ2021036 and ZDKJ2019011).

Q.J.: Conceptualization, Methodology, and Writing—original draft. H.J.: Conceptualization, Supervision, Writing—review and editing, and Funding acquisition. X.L.: Methodology and Writing—review and editing. Y.W.: Methodology and Writing—review and editing. I.-M.C.: Methodology and Writing—review and editing. L.H.: Methodology and Writing—review and editing. H.D.: Methodology and Writing—review and editing. Y.N.: Methodology and Writing—review and editing. D.G.: Methodology and Writing—review and editing.

## AUTHOR AFFILIATIONS

[1]CAS Key Laboratory for Experimental Study under Deep-sea Extreme Conditions, Institute of Deep-sea Science and Engineering, Chinese Academy of Sciences, Sanya, China
[2]University of Chinese Academy of Sciences, Beijing, China
[3]Southern Marine Science and Engineering Guangdong Laboratory, Zhuhai, Guangdong, China
[4]HKUST-CAS Sanya Joint Laboratory of Marine Science Research, Chinese Academy of Sciences, Sanya, China
[5]State Key Laboratory of Estuarine and Costal Research, East China Normal University, Shanghai, China

## AUTHOR ORCIDs

Qiuyun Jiang ⬥ http://orcid.org/0000-0003-3241-493X
Hongmei Jing ⬥ http://orcid.org/0000-0002-1795-5650
Hongpo Dong ⬥ http://orcid.org/0000-0003-4940-2943

## FUNDING

| Funder | Grant(s) | Author(s) |
|---|---|---|
| MOST | National Key Research and Development Program of China (NKPs) | 2022YFC2805505 | Hongmei Jing |
| MOST | National Key Research and Development Program of China (NKPs) | 2022YFC2805304 | Hongmei Jing |
| Hainan Province Science and Technology Special Fund | ZDKJ2021036 | Hongmei Jing |
| Hainan Province Science and Technology Special Fund | ZDKJ2019011 | Hongmei Jing |

## AUTHOR CONTRIBUTIONS

Qiuyun Jiang, Conceptualization, Investigation, Methodology, Writing – original draft | Hongmei Jing, Conceptualization, Funding acquisition, Supervision, Writing – review and editing | Xuegong Li, Methodology, Writing – review and editing | Ye Wan, Methodology,

Writing – review and editing | I-Ming Chou, Methodology, Writing – review and editing | Lijun Hou, Methodology, Writing – review and editing | Hongpo Dong, Methodology, Writing – review and editing | Yuhui Niu, Methodology, Writing – review and editing | Dengzhou Gao, Methodology, Writing – review and editing

## DATA AVAILABILITY

Sequences obtained were deposited in the National Center for Biotechnology Information (NCBI, https://www.ncbi.nlm.nih.gov) Sequence Read Archive (SRA) under the accession number PRJNA890167 for the bacterial 16S rRNA gene, PRJNA890198 for the archaeal 16S rRNA gene, PRJNA890329 for the *mcr*A gene, PRJNA890331 for the *pmo*A gene, and PRJNA890372 for the *dsr*B gene.

## ADDITIONAL FILES

The following material is available online.

### Supplemental Material

**Supplemental material (Spectrum02505-23-s0001.docx).** Tables S1 to S3; Fig. S1 to S4.

### Open Peer Review

**PEER REVIEW HISTORY (review-history.pdf).** An accounting of the reviewer comments and feedback.

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
