## [Reviewer comments · Microbiology Spectrum]

Microbiology Spectrum

Active pathways of anaerobic methane oxidization in deep-sea cold seeps of the South China Sea

Qiuyun Jiang, Hongmei Jing, Xuegong Li, Ye Wan, I-Ming Chou, Lijun Hou, Hongpo Dong, Yuhui Niu, and Deng-Zhou Gao

Corresponding Author(s): Hongmei Jing, Institute of Deep-sea Science and Engineering, Chinese Academy of Sciences, Sanya

Review Timeline:

Submission Date:	June 15, 2023
Editorial Decision:	September 13, 2023
Revision Received:	October 6, 2023
Accepted:	October 8, 2023

Editor: Jianjun Wang

Reviewer(s): Disclosure of reviewer identity is with reference to reviewer comments included in decision letter(s). The following individuals involved in review of your submission have agreed to reveal their identity: Ying Chen (Reviewer #1); Xiurong Xu (Reviewer #3)

Transaction Report:

DOI: <https://doi.org/10.1128/spectrum.02505-23>

September 13, 2023

Prof. hongmei jing
Sanya Institute of Deep-Sea Science and Engineering
No.28 Luhuitou road
Sanya
China

Re: Spectrum02505-23 (Active pathways of anaerobic methane oxidization in deep-sea cold seeps of the South China Sea)

Dear Prof. hongmei jing:

Thank you for submitting your manuscript to Microbiology Spectrum. Your manuscript has been seen by two reviewers. Please find their comments below for your revision. When submitting the revised version of your paper, please provide (1) point-by-point responses to the issues raised by the reviewers as file type "Response to Reviewers," not in your cover letter, and (2) a PDF file that indicates the changes from the original submission (by highlighting or underlining the changes) as file type "Marked Up Manuscript - For Review Only". Please use this link to submit your revised manuscript - we strongly recommend that you submit your paper within the next 60 days or reach out to me. Detailed instructions on submitting your revised paper are below.

Link Not Available

Sincerely,

Jianjun Wang

Journals Department
Reviewer comments:

Reviewer #1 (Comments for the Author):

My main concern about this manuscript is the description of the potential AOM rate. I would suggest that the authors should reorganize both parts of 3.1 and 4.1, and the quality of the figures needs to be improved.
line 246-247, It's hard for me to come to that conclusion that "the production of $^{13}\text{CO}_2$ in all sediments amended with different electron acceptors was always higher in the cold seeps than that in the trough during".
As shown in Fig.2 (SQ261, SQ82 and SQ81) that the highest observed values of $^{13}\text{CO}_2$ of " $^{13}\text{CH}_4$ " is higher than those of the groups with nitrate and nitrite. And according to the authors' statement on the calculation of the potential AOM rate, the potential AOM rate of " $^{13}\text{CH}_4$ " in samples SQ261, SQ82 and SQ81 should also be higher than those of the experimental groups with added nitrate and nitrite. However Fig.3 shows the opposite result.
It is not surprising that the experimental group " $^{13}\text{CH}_4$ " without any added electron acceptors showed a potential AOM rate during incubation, since some of the substances originally contained in the sediment samples, such as humic acid, can act as

electron acceptors for AOM.

Reviewer #3 (Comments for the Author):

- 1.Line 145: For the determination of $^{13}\text{CO}_2$ from $^{13}\text{CH}_4$, a blank control (which was added with equal of $^{12}\text{CH}_4$) should be designed.
- 2.Line 220: please provide a reference for the determination method or a brief procedure for the determination.
- 3.The statistical analysis of the data should be described in detail.
- 4.Line 225:Please provide a link to the web page where the data is deposited.

Staff Comments:

Preparing Revision Guidelines

Please return the manuscript within 60 days; if you cannot complete the modification within this time period, please contact me. If you do not wish to modify the manuscript and prefer to submit it to another journal, please notify me of your decision immediately so that the manuscript may be formally withdrawn from consideration by Microbiology Spectrum.

Dear editor and reviewers,

We are sincerely grateful for your comments and modified throughout the text accordingly. Please see below the details responds.

Reviewer #1 (Comments for the Author):

My main concern about this manuscript is the description of the potential AOM rate. I would suggest that the authors should reorganize both parts of 3.1 and 4.1, and the quality of the figures needs to be improved.

Response: According to the reviewer's suggestion, the description of the potential AOM rate has been re-organized in parts of 3.2 and 4.1 of the revised manuscript. In addition, the figures were compressed with reduced visualization in the previous version. All the figures have been checked and improved with higher resolution in the revised manuscript.

line 246-247, It's hard for me to come to that conclusion that "the production of $^{13}\text{CO}_2$ in all sediments amended with different electron acceptors was always higher in the cold seeps than that in the trough during".

Response: The average production of $^{13}\text{CO}_2$ in the Haima cold seeps (SQ56, SQ58, SQ81) and Site F cold seeps (SQ63, SQ261, SQ264) after 14-day incubation was 3.69 nmol g⁻¹ [wet sediment] and 6.20 nmol g⁻¹ [wet sediment], respectively. While in the trough (SQ82, SQ84, SQ87), it was 3.53 nmol g⁻¹ [wet sediment]. Based on the above results, we come to this conclusion. The above information was added in lines 256-259 of the revised manuscript.

As shown in Fig.2 (SQ261, SQ82 and SQ81) that the highest observed values of $^{13}\text{CO}_2$ of " $^{13}\text{CH}_4$ " is higher than those of the groups with nitrate and nitrite. And according to the authors' statement on the calculation of the potential AOM rate, the potential AOM rate of " $^{13}\text{CH}_4$ " in samples SQ261, SQ82 and SQ81 should also be higher than those of the experimental groups with added nitrate and nitrite. However Fig.3 shows the opposite result.

Response: In Fig. 2, $^{13}\text{CO}_2$ production in slurries of sediment amended with $^{13}\text{CH}_4$ and different electron acceptors during 14-day incubations was shown. While in Fig. 3 potential AOM rate in different samples and regions was shown. The potential AOM rate calculation was based on the $^{13}\text{CO}_2$ production from D3 to D7 of cultivation. Because the production of $^{13}\text{CO}_2$ increased rapidly from D3 to D7 during the incubation with various electron acceptors addition, and obvious consumption of nitrite appeared after 3-day incubation (Fig. 4A). And the potential AOM rate of " $^{13}\text{CH}_4$ " in sample SQ261 was higher than that of the experimental groups with added nitrate. The potential AOM rate of " $^{13}\text{CH}_4$ " in sample SQ81 was higher than that of the experimental groups with added nitrite and sulfate. The potential AOM rate of " $^{13}\text{CH}_4$ " in sample SQ82 was lower than that of the experimental groups with added nitrate, nitrite and sulfate. According to above information, Figs. 2 and 3 in fact showed consistent patterns.

It is not surprising that the experimental group " $^{13}\text{CH}_4$ " without any added electron acceptors showed a potential AOM rate during incubation, since some of the substances originally contained in the sediment samples, such as humic acid, can act as electron acceptors for AOM.

Response: Yes, the reviewer is correct. We admit that some of the substances originally contained in the sediment samples (such as humic acid, iron, manganese and etc.) may act as electron acceptors for AOM.

Reviewer #3 (Comments for the Author):

1 .Line 145: For the determination of $^{13}\text{CO}_2$ from $^{13}\text{CH}_4$, a blank control (which was added with equal of $^{12}\text{CH}_4$) should be designed.

Response: This study focused on the microbial communities and rates of SAMO and DAMO processes, and to elucidate the utilization of different electron acceptors (nitrite, nitrate and sulfate). The types of electron acceptors were independent variable, so " $^{13}\text{CH}_4$ " was used as the control group, while " $^{12}\text{CH}_4$ "

control group was not set up. This is consistent with the previous studies (Shen et al., 2016; Chen et al. 2021). In addition, the $^{13}\text{CO}_2$ produced in the original background would be present in each treatment group (different electron acceptors) as well, thus should have no effect on the trend in each treatment group.

Refs:

Shen, L. D., Hu, B. L., Liu, S., Chai, X. P., He, Z. F., Ren, H. X., Liu, Y., Geng, S., Wang, W., Tang, J. L., Wang, Y. M., Lou, L. P., Xu, X. Y., Zheng, P., 2016. Anaerobic methane oxidation coupled to nitrite reduction can be a potential methane sink in coastal environments. *Appl Microbiol Biotechnol* 100: 7171-7180. <https://doi.org/10.1007/s00253-016-7627-0>

Chen, F. Y., Zheng, Y. L., Hou, L. J., Niu, Y. H., Gao, D. Z., An, Z. R., Zhou, J., Yin, G. Y., Dong, H. P., Han, P., Liang, X., Liu, M., 2021. Microbial abundance and activity of nitrite/nitrate-dependent anaerobic methane oxidizers in estuarine and intertidal wetlands: Heterogeneity and driving factors. *Water Res.* 190: 116737. <https://doi.org/10.1016/j.watres.2020.116737>

2. Line 220: please provide a reference for the determination method or a brief procedure for the determination.

Responds: Thanks for the suggestion. The brief procedure for the determination was added in the lines 219-223 in the revised manuscript. A reference for the determination method was added in the line 225 as well.

3. The statistical analysis of the data should be described in detail.

Responds: Thanks for the suggestion. The statistical analysis of the data has been described in detail in the line 228-236 of the revised manuscript.

4. Line 225: Please provide a link to the web page where the data is deposited.

Response: The link to the web page where the data was deposited has been added in the line 239 of the revised manuscript. Thanks for the suggestion.

October 8, 2023

Prof. hongmei jing
Institute of Deep-sea Science and Engineering, Chinese Academy of Sciences, Sanya
No.28 Luhuitou road
Sanya
China

Re: Spectrum02505-23R1 (Active pathways of anaerobic methane oxidization in deep-sea cold seeps of the South China Sea)

Dear Prof. hongmei jing:

Thank you for your great efforts in revising the manuscript according to the comments of two reviewers. I found your revision is satisfying. Your manuscript has been accepted, and I am forwarding it to the ASM Journals Department for publication. You will be notified when your proofs are ready to be viewed.

Sincerely,

Jianjun Wang
Editor, Microbiology Spectrum
